environmental science/climatology/complexity

tipping point, critical threshold, hysteresis, tipping cascade, Earth system, eutrophication

**Author for correspondence:**
Jonathan F. Donges
e-mail: donges@pik-potsdam.de

# Emergence of cascading dynamics in interacting tipping elements of ecology and climate

Ann Kristin Klose[1,2], Volker Karle[1,3],
Ricarda Winkelmann[1,4] and Jonathan F. Donges[1,5]

[1]Earth System Analysis, Potsdam Institute for Climate Impact Research, Member of the Leibniz Association, Telegrafenberg A31, 14473 Potsdam, Germany
[2]Carl von Ossietzky University Oldenburg, Oldenburg, Germany
[3]Institute of Science and Technology Austria, Am Campus 1, 3400 Klosterneuburg, Austria
[4]Department of Physics and Astronomy, University of Potsdam, 14469 Potsdam, Germany
[5]Stockholm Resilience Centre, Stockholm University, 10691 Stockholm, Sweden

JFD, 0000-0001-5233-7703

In ecology, climate and other fields, (sub)systems have been identified that can transition into a qualitatively different state when a critical threshold or tipping point in a driving process is crossed. An understanding of those tipping elements is of great interest given the increasing influence of humans on the biophysical Earth system. Complex interactions exist between tipping elements, e.g. physical mechanisms connect subsystems of the climate system. Based on earlier work on such coupled nonlinear systems, we systematically assessed the qualitative long-term behaviour of interacting tipping elements. We developed an understanding of the consequences of interactions on the tipping behaviour allowing for tipping cascades to emerge under certain conditions. The (narrative) application of these qualitative results to real-world examples of interacting tipping elements indicates that tipping cascades with profound consequences may occur: the interacting Greenland ice sheet and thermohaline ocean circulation might tip before the tipping points of the isolated subsystems are crossed. The eutrophication of the first lake in a lake chain might propagate through the following lakes without a crossing of their individual critical nutrient input levels. The possibility of emerging cascading tipping dynamics calls for the development of a unified theory of interacting tipping elements and the quantitative analysis of interacting real-world tipping elements.

# 1. Introduction

Many natural systems exhibit nonlinear dynamics and can undergo a transition into a qualitatively different state when a critical threshold is crossed. Those systems are called tipping elements and the corresponding threshold in terms of a critical parameter is the tipping point of the system. A precise mathematical definition is given in [1]. Examples for tipping elements can be found in ecology as a specific type of regime shifts [2,3] such as the transition of a shallow lake from a clear to a turbid state [4–9]. Furthermore, subsystems of the Earth system [1,10,11] such as the thermohaline circulation [12–18] or the Greenland ice sheet [19,20] have been identified as tipping elements.

The term tipping point among other roots originated from describing the changing prevalence of ethnically diverse population in a US community [21–25] and has been applied to natural systems more recently. However, the idea that systems may show such nonlinear behaviour has already been developed within the frameworks of dynamical systems and catastrophe theory [26–30]. The latter theory received increasing attention and has been applied to several real-world systems in the period after its introduction [31]. Its extensive use has been criticized [32–35] so that it became a mathematical theory without much recent influence [36]. Mostly independently of the results given by catastrophe theory, critical transitions, tipping points and regime shifts have been analysed in ecology [2,9,37–39] using the concepts of multistability and resilience [40,41]. Some first attempts to define a climatic tipping element relating to abrupt climate shifts can be found in [42–44].

Different types of tipping points are discussed in the literature [9,23,37–39,45]. First, a qualitative change of the system's state when a continuously changing control parameter crosses a threshold is called bifurcation-induced tipping [23,46–48]. Noise can induce a transition into an alternative stable state without a change of the system's control parameter [23,38,49]. Furthermore, rate-induced tipping describes the shift to a qualitatively different state when the rate of change of a control parameter crosses a critical threshold [23,45,49,50].

It is known that bifurcation-induced tipping, even though often mentioned, is not the only possible type of tipping [24,45,49,51]. Nevertheless, the response of many natural systems to a control parameter can be described in terms of a double fold bifurcation [48,52,53].

Real-world tipping elements are not independent from each other [52], but there may exist complex interactions between them. Potential interactions through various physical mechanisms were revealed for tipping elements in the climate system [54]. As an example, meltwater influx into the North Atlantic as a result of a tipping of the Greenland ice sheet could weaken the Atlantic meridional overturning circulation [55]. Lake chains or rivers can be seen as an ecological example for coupled tipping elements. Each lake or river section in the chain can undergo a transition from a clear to a turbid state in response to nutrient input [5,6,8]. The single lakes can in reality be connected through small rivers or streams and can therefore not be considered independently [56–58].

The tipping probability of a certain tipping element might be influenced by the behaviour of other interacting tipping elements [54,59]. As a consequence, crossing of a critical threshold of a first tipping element could trigger, as a domino effect, a critical transition in a coupled tipping element or even tipping cascades [37,59–64]. In the following, we use the term tipping cascade for a critical transition triggered by a preceding tipping event of an influencing system. The heterogeneity of the subsystems as well as the coupling strength may be important factors that influence the overall system behaviour [65] and should be considered in the analysis of coupled tipping elements. In the case of interacting climate tipping elements, a tipping cascade may impose a considerable risk on human societies [66].

Different attempts to analyse the influence of coupling between different tipping elements on their tipping behaviour have been followed. The development of critical transitions in lake chains was studied using established models of lake eutrophication [58,67,68]. In analogy to wave propagation in discrete media [69–72], the spread of local disturbances in spatially extended, bistable ecosystems was analysed for explicit ecological examples [73] and more theoretically [74]. In addition, cascades may occur on networks [75–78] and networks of networks [79–83]. [52,84,85] analysed the system behaviour of special cases of coupled cusp catastrophes. The possible appearance of tipping cascades in coupled bifurcational systems and, in particular, in the climate system was supported by results from coupling conceptual models of the Atlantic meridional overturning circulation and El Niño-Southern Oscillation [86].

Consequences of interactions between tipping elements, their nonlinear dynamics as well as the possible development of tipping cascades in systems of interacting tipping elements have not been assessed systematically so far. Here, we make an advance in the theory of interacting tipping elements

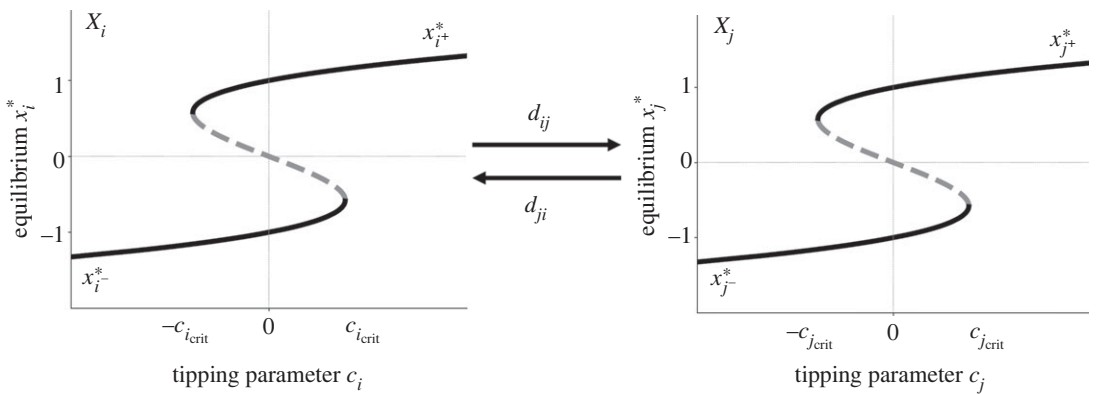

**Figure 1.** Schematic coupled tipping elements. Two subsystems $X_i$ and $X_j$ are coupled via the coupling functions $C_i$ and $C_j$ with coupling strengths $d_{ji}$ and $d_{ij}$. The dynamics of both subsystems is given by the normal form of the cusp catastophe. Shown are the equilibria $x_i^*$, $x_j^*$ depending on the tipping parameter $c_i$, $c_j$.

focusing on bifurcation-induced tipping in the form of cusp catastrophes. The special cases presented in [52,85] were extended by looking at uni- and bi-directional coupling and tipping chains consisting of two and three elements. We explored the tipping behaviour of the interacting system extensively under the influence of different coupling types (positive and negative interactions) and the coupling strength and particularly focused on identifying conditions that favour tipping cascades. In addition, we applied our theoretical results to real-world systems to reveal mathematically possible tipping cascades in ecological systems such as lake chains and in the climate system.

## 2. Model

We use a conceptual model of tipping elements in order to investigate the qualitative long-term behaviour of coupled subsystems each of which exhibits critical transitions. In particular, we consider a continuous dynamical system $\dot{x}_i(t) = f_i(x_1, \ldots, x_n)$ in $n$-dimensions, where each component $x_i(t) \in \mathbb{R}$ corresponds to a generic tipping element $X_i$.

The dynamics of the tipping elements is modelled with the topological normal form of the cusp bifurcation [52,85], i.e. the most generic polynomial system exhibiting this type of tipping behaviour (figure 1). The long-term behaviour of many real-world systems in terms of the system's state, such as the strength of the thermohaline circulation [12,15,17,18], the ice thickness of the Greenland ice sheet [19] and the algae density in shallow lakes [6,8], is qualitatively represented by a slice of the cusp catastrophe, a double fold bifurcation, showing bistability, hysteresis properties and transitions to a qualitatively different state when a critical threshold is crossed [87]. In contrast to other possible bifurcations such as the transcritical, pitchfork or Hopf bifurcations, the double fold bifurcation as a 'dangerous' bifurcation [46] captures the catastrophic nature of tipping which is of major interest here. A tipping element $X_i$ is then represented by

$$f_i^0(x_i) = a_i x_i(t) - b_i x_i^3(t) + c_i \quad \text{with } a_i, b_i, c_i \in \mathbb{R}, \tag{2.1}$$

where $a_i, b_i > 0$ and $f_i^0$ corresponds to the uncoupled case.

The parameter $a_i$ corresponds to the distance between the upper and lower layers of stable equilibria of the cusp and the parameter $b_i$ controls the strength of the nonlinearity in the system. Both parameters are fixed throughout the analysis and chosen to assure bistability of the subsystems for a certain range of the parameter $c_i$ allowing for tipping behaviour. The control parameter $c_i$ is then associated with a tipping parameter the size of which determines whether the system is in this bistable range or not. By leaving the bistability range, a critical transition from one stable state to another may arise (figure 1):

For $-c_{i_{\text{crit}}} < c_i < c_{i_{\text{crit}}}$, equation (2.1) has one negative stable equilibrium $x_{i-}^* < 0$ and a positive stable equilibrium $x_{i+}^* > 0$ as alternative stable states. We call $x_{i-}^* < 0$ and $x_{i+}^* > 0$ the normal and the alternative state, respectively. Increasing the control parameter $c_i$ such that the threshold $c_{i_{\text{crit}}}$ is crossed, the normal state $x_{i-}^*$ disappears and only the alternative state $x_{i+}^*$ exists. If the system resided in the normal state $x_{i-}^*$, it transitions to the alternative state $x_{i+}^*$ for $c_i > c_{i_{\text{crit}}}$. Analogously, for $c_i < -c_{i_{\text{crit}}}$, only the normal state $x_{i-}^*$ exists and for lowering the control parameter $c_i$ below $-c_{i_{\text{crit}}}$ the system falls

**Table 1.** Overview of linearly coupled tipping elements studied in the literature.

| reference | coupling type | parameter choices |
|---|---|---|
| | $n = 2$ | |
| [52] | master–slave system | $b_1 = b_2 = 1$ |
| | with linear coupling | $a_1 = a_2 = 1$ |
| | | $C_1(x_1, x_2) = 0$ |
| | | $C_2(x_1, x_2) = dx_1$ |
| [85] | Kadyrov style | $a_1 = a_2 = 1$ |
| | | $b_1 = b_2 = 1$ |
| | | $c_1 = c_2 = 0$ |
| | | $C_1(x_1, x_2) = d_{21}x_2$ |
| | | $C_2(x_1, x_2) = d_{12}x_1$ |
| | | symmetric coupling: $d_{21} = d_{12}$ |
| | | asymmetric coupling: $d_{21} \neq d_{12}$ |
| | $n = 3$ | |
| [52] | master–slave–slave system | $b_1 = b_2 = b_3 = 1$ |
| | with linear coupling | $a_1 = a_2 = a_3 = 1$ |
| | | $C_1(x_1, x_2, x_3) = 0$ |
| | | $C_2(x_1, x_2, x_3) = dx_1$ |
| | | $C_3(x_1, x_2, x_3) = dx_2$ |
| | $n > 2$ | |
| [84] | $n$ equations | $x_i = -x_i^3 + A_{ij}x_j$ |
| | coupled in a graph | $A_{ij}$: matrix of size $N \times N$ |

from the disappearing alternative state $x_{i+}^*$ to the normal state $x_{i-}^*$ (figure 1). The transition of the uncoupled tipping elements through a changing control parameter $c_i$ at the critical manifold [88] given by the roots of the polynomial can be quantified: depending on the sign of the discriminant $D_i^0 = (b_i c_i/2)^2 - b_i(a_i/3)^3$ there are either one ($D_i^0 > 0$) or two ($D_i^0 \leq 0$) stable equilibria determined by $f_i^0(x_i^*) = 0$.

For given $a_i$ and $b_i$ and setting $D_i^0 = 0$, the critical value for the control parameter $c_{i_{\mathrm{crit}}}(a_i, b_i) = \pm 2\sqrt{\frac{1}{b_i}(\frac{a_i}{3})^3}$ can be calculated, where the transition into a regime with only one equilibrium takes place. We call $c_{i_{\mathrm{crit}}}(a_i, b_i)$ the intrinsic tipping points for an uncoupled tipping element as given by equation (2.1).

In the following, we couple the subsystems with each other using a coupling function $C_i \in \mathbb{R}$ (figure 1). Subsystem $X_i$ then becomes

$$f_i(x_1, x_2, \ldots, x_n) = a_i x_i(t) - b_i x_i^3(t) + c_i + C_i(x_1(t), x_2(t), \ldots, x_n(t)), \tag{2.2}$$

with $a_i, b_i > 0$.

For simplicity, we choose a linear coupling [52,85]. The linear coupling function for subsystem $X_i$ then reads as

$$C_i(x_1(t), x_2(t), \ldots, x_n(t)) = \sum_{j=1}^n d_{ji} x_j(t) \quad \text{with } i \neq j, \tag{2.3}$$

where a coupling $d_{ji} \neq 0$ indicates an influence of another subsystem $X_j$ to subsystem $X_i$. Even though equations (2.2) and (2.3) provide the simplest equations to describe the threshold behaviour of $n$ coupled tipping elements, they can be used for understanding the qualitative features of all systems with the same critical behaviour. Using the concept of topological equivalence [88], the critical behaviour of a class of more complicated real-world systems can be mapped to the system above. Table 1 provides an overview of special cases of coupling between interacting cusp catastrophes

investigated in the literature [52,84,85]. In addition, more theoretical analyses on bifurcations of coupled cell systems (with symmetry properties) have been conducted (e.g. [89]).

For $n = 2$, the corresponding equations read

$$\dot{x}_1(t) = a_1 x_1(t) - b_1 x_1^3(t) + c_1 + d_{21} x_2(t)$$

and
$$\dot{x}_2(t) = a_2 x_2(t) - b_2 x_2^3(t) + c_2 + d_{12} x_1(t). \tag{2.4}$$

with $a_i$, $b_i > 0$. With $d_{21} = 0$ and $d_{12} \neq 0$, we recover a generic master–slave configuration. The stable equilibria can be determined analogously to the uncoupled case with $f_i(x_1^*, x_2^*, \ldots, x_n^*) = 0 \ \forall i$. The discriminant for the second tipping element $X_2$ becomes $D_2 = (b_2 \, (c_2 + d_{12} x_1^*)/2)^2 - b_2 \, (a_2/3)^3$.

Note that $D_2$ is a function of the control parameter $c_2$, the coupling strength $d_2$ and the equilibrium $x_1^*$. The number of stable equilibria of subsystem $X_2$ depends on the sign of the discriminant. For $D_2 \leq 0$, we find two stable equilibria and for $D_2 > 0$, we find one stable equilibrium. The threshold of the control parameter $c_2$ at which the number of solutions changes is obtained by solving $D_2 = 0$ and is given by

$$c_2 = -d_{12} x_1^* \pm c_{2_{\text{crit}}}(a_2, b_2), \tag{2.5}$$

where

$$c_{2_{\text{crit}}}(a_2, b_2) = 2\sqrt{\frac{1}{b_2}\left(\frac{a_2}{3}\right)^3}, \tag{2.6}$$

as the effective tipping point of the coupled tipping element $X_2$.

In the following section, we will elaborate on how one can infer the qualitative behaviour of the coupled system using this expression.

# 3. Results

Different types of tipping behaviour of a coupled system can be derived for the governing system of equations (2.4). For simplicity, let us consider the case $a_i = 1$, $b_i = 1$ (arbitrary $b_i$ can be achieved by rescaling $x_i$) for $i = 1, 2$ here and thereafter and $d_{21} = 0$, i.e. unidirectional coupling. Subsystem $X_2$ leaves the bistable range for

$$c_2 + d_{12} x_1^* \geq c_{2_{\text{crit}}}(a_2, b_2), \tag{3.1}$$

following expression (2.5) for the effective tipping point of the coupled tipping element $X_2$ in the previous §2, possibly giving rise to a critical transition to the alternative state $x_{2+}^* > 0$.

Based on this simple system, rules on the spread of tipping processes in the considered system of coupled tipping elements are formulated in the following. These tipping rules depend on the type of coupling, i.e. whether the subsystems are positively or negatively coupled, as well as on the relation between the control parameters $c_1$ (determining the possible stable states of subsystem $X_1$) and $c_2$ and the absolute value of the coupling strength $d_{12}$.

Let $d_{12} > 0$, i.e. subsystem $X_2$ is positively coupled to subsystem $X_1$. Then:

— **Facilitated tipping** (figure 2, upper panel): Assume that subsystem $X_1$ is in its alternative state $x_{1+}^*$. Note that this assumption can be fulfilled if either subsystem $X_1$ transitions to the alternative state by crossing its intrinsic tipping point $c_{1_{\text{crit}}}$ with an increase of the control parameter $c_1$ or if subsystem $X_1$ simply occupies the alternative state (which is in general possible for $c_1 > -c_{1_{\text{crit}}}$). Then subsystem $X_2$ is pushed towards its tipping point in our model and can undergo a critical transition to its alternative state $x_{2+}^*$ for $c_2 \geq c_{2_{\text{crit}}} - d_{12} x_1^*$. The effective tipping point of subsystem $X_2$ is lower than its intrinsic tipping point $c_{2_{\text{crit}}}$. The higher the coupling strength, the lower the necessary critical value of the control parameter $c_2$ for which subsystem $X_2$ can tip.
— **Impeded tipping** (figure 2, lower panel): If subsystem $X_1$ is in its normal state $x_{1-}^*$, subsystem $X_2$ is pulled away from its tipping point in our model and can undergo a critical transition for $c_2 \geq c_{2_{\text{crit}}} + d_{12}|x_1^*|$. The effective tipping point of subsystem $X_2$ is higher than its intrinsic tipping point $c_{2_{\text{crit}}}$. The higher the coupling strength, the higher the necessary critical value of the control parameter $c_2$ for which subsystem $X_2$ can tip.
— **Back-tipping**: Assume that subsystem $X_1$ is in its normal state $x_{1-}^*$. If subsystem $X_2$ occupies the alternative state $x_{2+}^*$, subsystem $X_2$ can tip back to the normal state $x_{2-}^*$ for $c_2 < -c_{2_{\text{crit}}} + d_{12}|x_1^*|$ (figure 3, upper panel). This behaviour especially occurs for a high coupling strength $d_{12}$ and

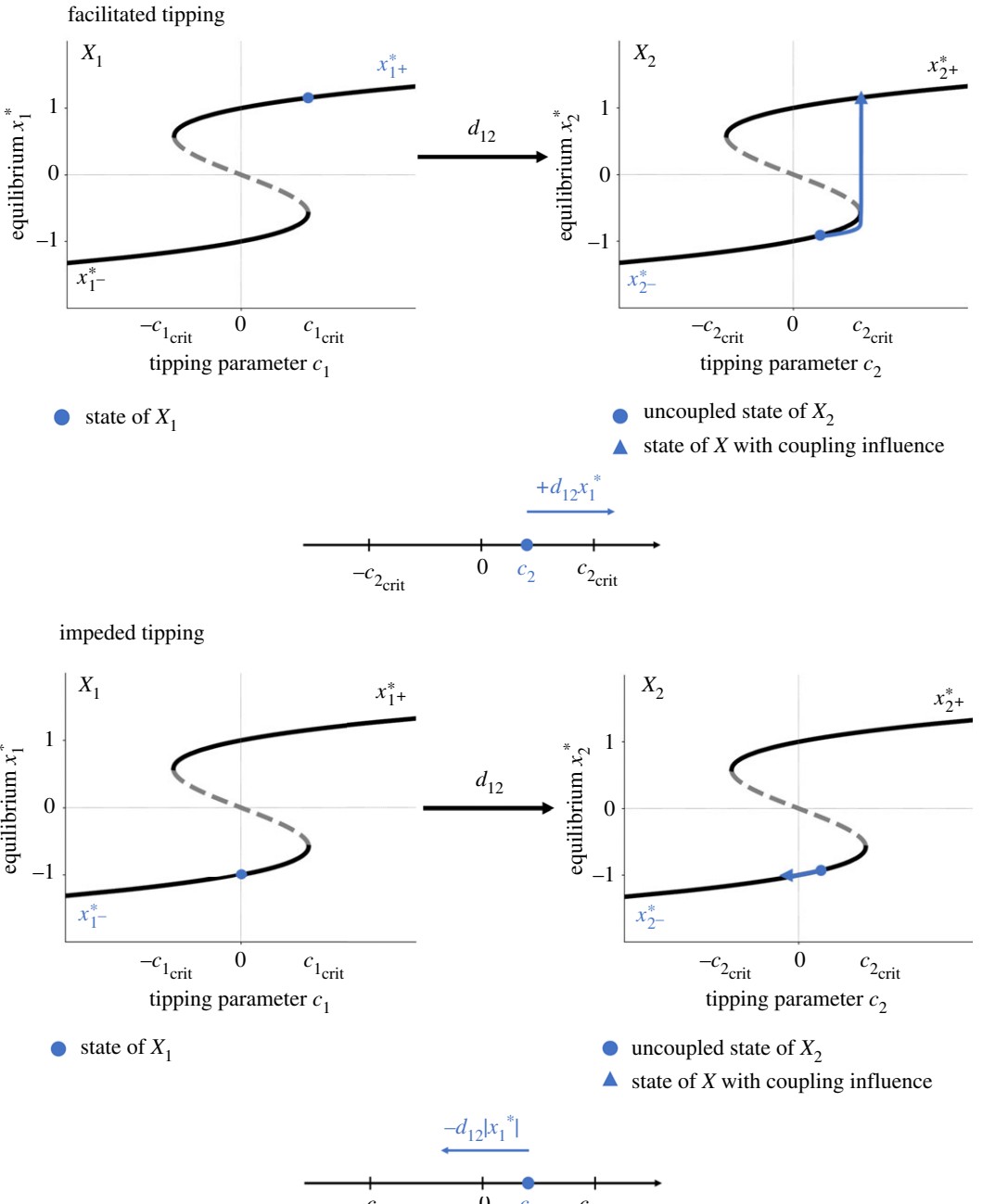

**Figure 2.** Schematic of the tipping rules of facilitated (upper panel) and impeded tipping (lower panel) for $d_{12} > 0$. The blue dot in the left bifurcation diagram represents a possible state of the master system $X_1$. The master system $X_1$ influences the slave system $X_2$ via a linear coupling with a coupling strength $d_{12} > 0$ and results in the shift of the uncoupled slave system's state (indicated by a blue dot in the right bifurcation diagram) along the blue line.

small values of the control parameter $c_2$. However, subsystem $X_2$ is staying in the alternative state $x_{2+}^*$ if $-c_{2_\text{crit}} + d_{12}|x_1^*| < c_2$. Here, subsystem $X_2$ is pushed into the bistable range of the system (figure 3, lower panel). This behaviour especially occurs for a high coupling strength $d_{12}$ and high values of the control parameter $c_2$.

Let $d_{12} < 0$, i.e. subsystem $X_2$ is negatively coupled to subsystem $X_1$. Then:

— **Impeded tipping**: Assume that subsystem $X_1$ is in its alternative state $x_{1+}^*$. Note that this assumption can be fulfilled if either subsystem $X_1$ transitions to the alternative state by crossing its intrinsic

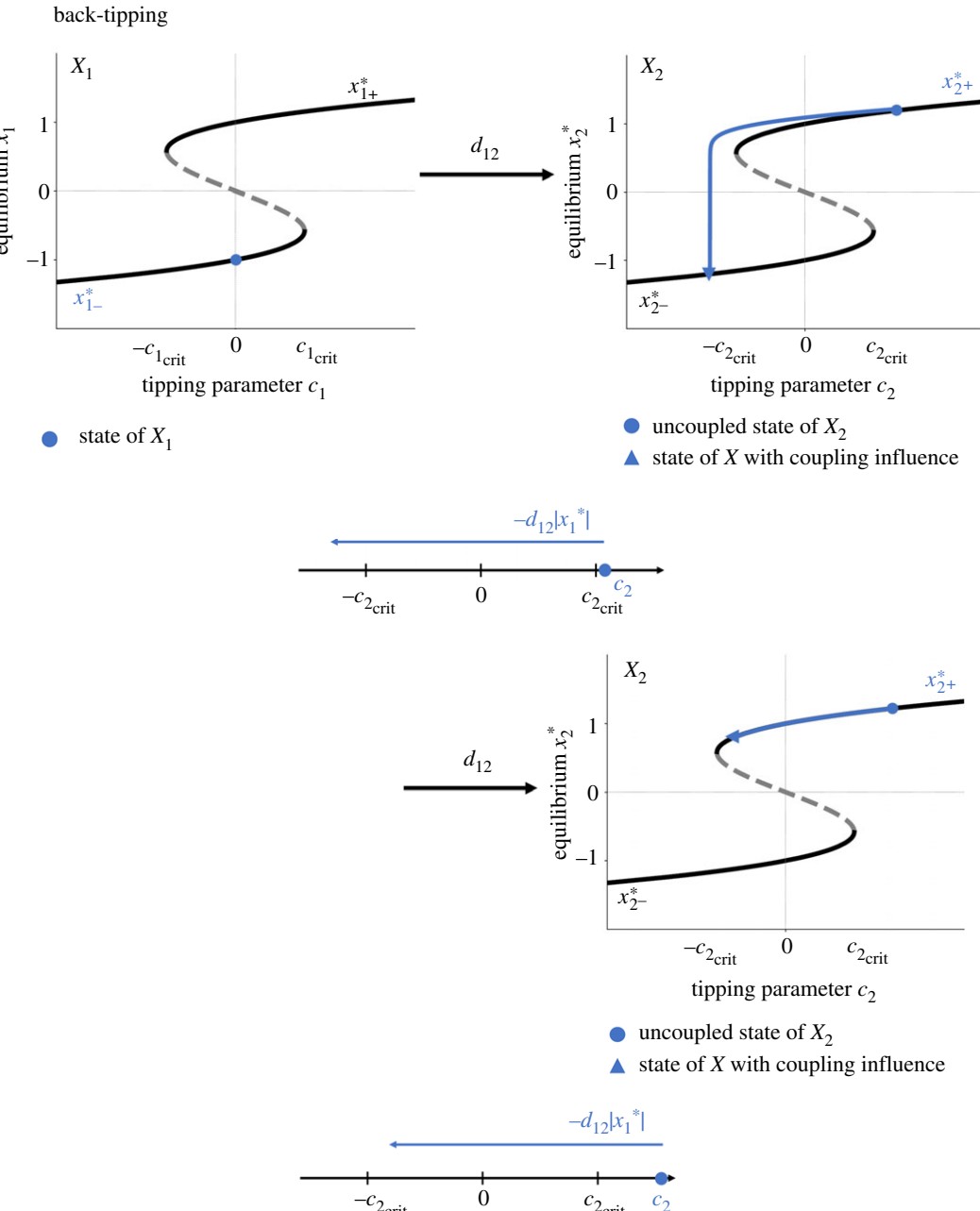

**Figure 3.** Schematic of the tipping rule of back-tipping for $d_{12} > 0$. The blue dot in the left bifurcation diagram represents a possible state of the master system $X_1$. The master system $X_1$ influences the slave system $X_2$ via a linear coupling with a coupling strength $d_{12} > 0$ and results in the shift of the uncoupled slave system's state (indicated by a blue dot in the right bifurcation diagram) along the blue line.

tipping point $c_{1_{\text{crit}}}$ with an increase of the control parameter $c_1$ or if subsystem $X_1$ simply occupies the alternative state (which is in general possible for $c_1 > -c_{1_{\text{crit}}}$). Subsystem $X_2$ is pulled away from its tipping point in our model and can undergo a critical transition for $c_2 \geq c_{2_{\text{crit}}} + |d_{12}\|x_1^*|$. The effective tipping point of subsystem $X_2$ is higher than its intrinsic tipping point $c_{2_{\text{crit}}}$. The higher the coupling strength, the higher the necessary critical value of the control parameter $c_2$ for which subsystem $X_2$ can tip.

— **Facilitated tipping**: If subsystem $X_1$ is in its normal state $x_{-}^*$, subsystem $X_2$ is pushed towards its tipping point in our model and can undergo a critical transition to its alternative state $x_{2+}^*$ for $c_2 \geq c_{2_{\text{crit}}} - |d_{12}\|x_1^*|$. The effective tipping point of subsystem $X_2$ is lower than its intrinsic tipping point $c_{2_{\text{crit}}}$. The higher the coupling strength, the lower the necessary critical value of the control parameter $c_2$ for which subsystem $X_2$ can tip.

— **Back-tipping**: Assume that subsystem $X_1$ is in its normal state $x_{1-}^*$ . If subsystem $X_2$ occupies the alternative state $x_{2+}^*$, subsystem $X_2$ can tip back to the normal state $x_{2-}^*$ if $c_2 < -c_{2_{crit}} - |d_{12}\|x_1^*|$. This behaviour especially occurs for a high coupling strength $d_{12}$ and a low value of the control parameter $c_2$. However, subsystem $X_2$ stays in the alternative state if $c_2 \geq -c_{2_{crit}} - |d_{12}\|x_1^*|$. Here, subsystem $X_2$ is pulled to the bistable area of the system. This behaviour especially occurs for a high coupling strength $|d_{12}|$ and high values of the control parameter $c_2$.

These tipping rules based on the analytic solution of two unidirectionally coupled tipping elements give an impression of the interplay between certain system parameters and their influence on the tipping process. Going beyond this, using numerical calculations of fixed points and their stability (via the eigenvalues of the system's Jacobian at the respective fixed point), the overall qualitative long-term behaviour of up to three interacting tipping elements with uni- and bi-directional coupling of varying sign has been assessed.

A stability map displays the number of stable equilibria of the system under consideration in the control parameter space for fixed coupling strengths. Multiple stability maps have been calculated for various combinations of the coupling strengths. The stability maps have been arranged in the form of a matrix, where one matrix element corresponds to one stability map. For illustrative purposes, we refer to the example given in figure 4 for two interacting tipping elements.

The system loses or gains stable fixed points through the variation of one or various control parameters $c_i$ for fixed coupling strengths, which is associated with switches between the areas of different number of stable equilibria in the control parameter space by crossing the boundaries between the coloured areas. In addition, the phase space portrait may change.

Depending on the changes in the phase space (in terms of the stable fixed points and the flow) and the occupied state of the system, different types of system behaviour emerge. Combining the stability maps with phase space portraits (example given in figure 5 for illustrative purposes), the different areas in the stability maps can be characterized in terms of the emerging system behaviour, and possible critical transitions can be identified. Depending on which state the system was in, critical transitions can occur or not. For example, if the system resided in an equilibrium which lost stability and disappeared through the variation of one (or multiple) control parameters, the flow in the phase space suggests the state to which the system may transition.

In the following, results for selected examples of coupled tipping elements are shown. First, a simple master–slave system with a unidirectional coupling is presented in example 3.1. In example 3.2, the previous system is extended by an additional negative coupling resulting in a bidirectionally coupled system of two tipping elements. Finally, the propagation in a unidirectionally coupled system consisting of three tipping elements is described in example 3.3. We only analyse the system behaviour for $c_i \geq 0$.

**Example 3.1.** Master–slave system for $d_{12} > 0$, e.g. a pair of lakes

The behaviour of a master–slave system ($n = 2$) given by equations (2.2) and (2.3) with $i = 1, 2$ and positive coupling $d_{12} > 0$ and $d_{21} = 0$, which was used for the derivation of the tipping rules, is described in more detail in the following. This type of coupling can be seen as an example for a pair of interacting lakes. Each lake may undergo a transition from a clear to a turbid state when some critical magnitude of the nutrient input as the tipping parameter is exceeded [6,8]. It is assumed that the lakes are connected [57,58,67] through a unidirectional water stream. Critical transitions can be derived using the numerically calculated phase portrait in combination with the stability map for a fixed coupling strength $d_{12} > 0$ (figure 5). Note that figure 5 is a zoom into figure 4 at the lower right. The system has four stable equilibria for small values of the control parameters $c_1$ and $c_2$ which are separated by four saddles and an unstable node in the centre of the phase space.

With increasing control parameter $c_1$, a critical transition of subsystem $X_1$ occurs in our model when its intrinsic tipping point $c_{1_{crit}}$ is crossed (figure 5, moving from lower left to lower right along the green arrow), given that the system occupied one of the stable equilibria which lose stability for $c_1 > c_{1_{crit}}$.

With increasing control parameter $c_2$, a critical transition of subsystem $X_2$ occurs in our model even if $c_2 < c_{2_{crit}}$, given that subsystem $X_1$ is in the alternative state (figure 5, moving upwards from the lower left along the yellow arrow). The coupled subsystem $X_2$ tips at an effective tipping point lower than its intrinsic tipping point $c_{2_{crit}}$.

For an increase of the control parameter $c_1$ above the intrinsic tipping point $c_{1_{crit}}$ and a slight increase of the control parameter $c_2$, a tipping cascade, starting from the normal states of $X_1$ and $X_2$, with a critical

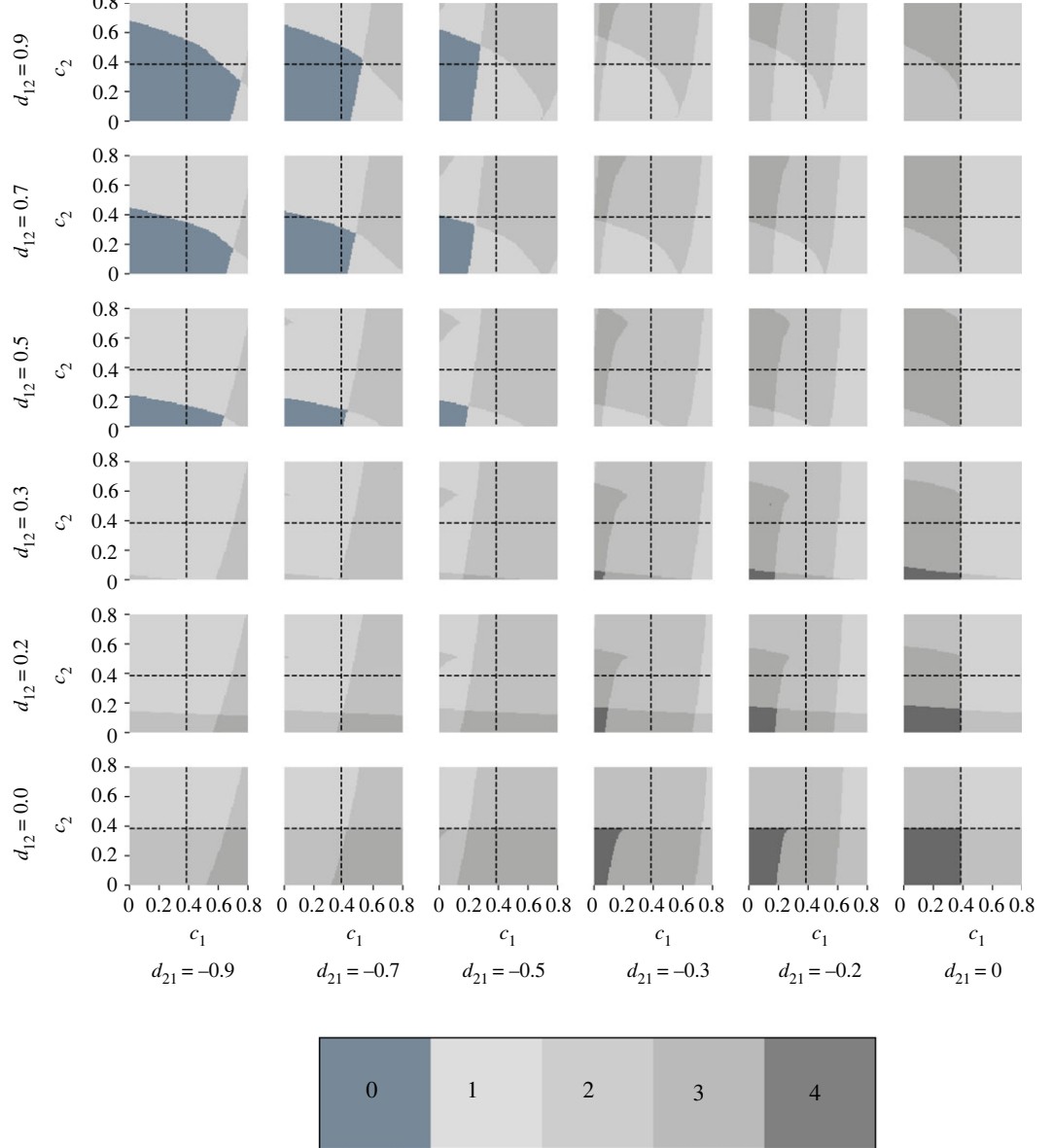

**Figure 4.** Number of stable fixed points of the system consisting of two bidirectionally coupled tipping elements depending on the control parameters $c_1$ and $c_2$ and the coupling strengths $d_{21} \leq 0$ and $d_{12} \geq 0$ in a matrix of stability maps. A stability map shows the number of stable fixed points in the $(c_1, c_2)$–space for a specific coupling strength, where a certain number of stable fixed points is associated with a specific colour. Note that different areas in the control parameter space with the same colour have the same number of stable fixed point but they do not necessarily have the same phase portrait. The dashed lines represent the intrinsic tipping point of the respective subsystem. The position of a stability map in the matrix is determined by the coupling strength. In the blue–grey region for high coupling strengths with opposite sign but same magnitude indicating the absence of stable equilibria a stable limit cycle (Kadyrov oscillations in [85]) can be observed.

transition in subsystem $X_1$ and a following transition in subsystem $X_2$ arises in our model (figure 5, moving from the lower left to the right column along the pink arrow). Note that the cascade occurs before the intrinsic tipping point of subsystem $X_2$ is crossed.

There is a change in the system behaviour for an increasing coupling strength (electronic supplementary material, figure S1). The previously described area with only one stable fixed point of the two subsystems in the alternative state for $c_1 > c_{1_{\mathrm{crit}}}$ exists for extremely low values of $c_2 \ll c_{2_{\mathrm{crit}}}$. Therefore, a tipping cascade can occur for even lower values of the control parameter $c_2 \ll c_{2_{\mathrm{crit}}}$ than for a system with lower coupling strength. For $c_1 < c_{1_{\mathrm{crit}}}$ and low values of the control parameter $c_2$,

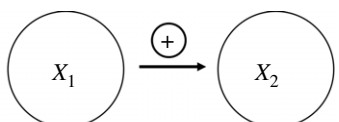

**Figure 5.** Number of stable fixed points and phase space portraits in a master–slave system with a low positive coupling strength $d_{12} = 0.2 > 0$ depending on the control parameters $c_1$ and $c_2$. The dashed lines represent the intrinsic tipping point of the respective subsystem. The phase space portraits allow to derive the possible critical transitions in the master–slave system. Within the phase space portraits stable fixed points are shown in orange, while unstable fixed points are shown in red. The background colour indicates the normalized speed $v = \sqrt{\dot{x}_1^2 + \dot{x}_2^2}/v_{\max}$ going from close to zero (purple) to fast (yellow–green).

subsystem $X_2$ either transitions to the alternative state for $c_2 < c_{2_{\text{crit}}}$, given that subsystem $X_1$ is in the alternative state, or subsystem $X_2$ tips back from the alternative state to the normal state, given that subsystem $X_1$ has not tipped. For $c_1 < c_{1_{\text{crit}}}$ and an increased control parameter $c_2$, the critical transition of subsystem $X_2$ to the alternative state, given that subsystem $X_1$ occupies the alternative state, is the only transition that can be observed.

**Example 3.2.** Bidirectional interaction of two tipping elements, e.g. Greenland ice sheet and Atlantic meridional overturning circulation

Consider a system consisting of two ($n = 2$) tipping elements given by equations (2.2) and (2.3) with $i = 1, 2$ and a bidirectional coupling where $d_{21} < 0$ and $d_{12} > 0$. This type of coupling can for instance be found in the interaction of the Greenland ice sheet (GIS) and the Atlantic meridional overturning circulation (AMOC), whose long-term behaviour may be represented by a double fold as suggested by (simple) models [14,17,90]: increased meltwater influx into the North Atlantic due to tipping of the GIS could lead to a weakening or even shutdown (tipping) of the AMOC [55], i.e. introducing a positive coupling. At the same time, a slowdown of the AMOC leads to a relative cooling around Greenland and hence corresponds to a negative coupling [54]. The system is analysed for a low and a high coupling strength (where $d_{21}$ and $d_{12}$ have opposite signs but the same magnitude) due to a substantial change of the qualitative behaviour towards higher coupling strength. There currently is not sufficient knowledge on the strength of the interaction between the GIS and the AMOC so that neither a low nor a high coupling strength can be excluded for certain. The number of stable equilibria and possible critical transitions for different parameter settings are given in figure 6 as zoom into figure 4 for low coupling strengths (lower right). A starting point of the analysis of the system behaviour for the low coupling strength is the area of four stable equilibria for low values of the control parameters $c_1$ and $c_2$ in figure 6.

With increasing control parameter $c_1$, a critical transition of the GIS as subsystem $X_1$ is possible for $c_1 < c_{1_{crit}}$ in our model, given that the AMOC as subsystem $X_2$ is in the normal state (figure 6, moving from lower left to the right along the yellow arrow). The GIS might tip at an effective tipping point which is lower than the intrinsic tipping point of the isolated subsystem. A critical transition of the GIS for the AMOC being in the alternative state is possible with a further increase of the control parameter $c_1$ for $c_1 \gg c_{1_{crit}}$ (figure 6, moving from lower left to the right along the second yellow arrow).

With increasing control parameter $c_2$, a critical transition of the AMOC as subsystem $X_2$ to a state with weakened strength (i.e. the alternative state) is possible in our model for $c_2 < c_{2_{crit}}$, given that the GIS as subsystem $X_1$ has already tipped (figure 6, moving from lower left upwards along the yellow arrow). The AMOC might tip at an effective tipping point which is lower than the intrinsic tipping point of the isolated subsystem. Given that the GIS is in its normal state, a critical transition of the AMOC is possible with a further increase of the control parameter $c_2$ for $c_2 \gg c_{2_{crit}}$ at an effective tipping point higher than its intrinsic tipping point (figure 6, moving from lower left upwards along the second yellow arrow). As a result of the model formulation, the GIS would still pull the thermohaline circulation (THC) away from its tipping point even though it has already started to melt but has not tipped to $x_{1+}^*$ (i.e. $0 > x_1^* > -1$). It would be possible to adjust the coupling function or the dynamics of each tipping element (e.g. [76]) so that already a slight change of the GIS state towards the alternative state without a complete critical transition to full loss of the ice sheet would push the AMOC towards its own tipping point.

For a slight increase of both control parameters $c_1$ and $c_2$, a critical transition of the GIS as well as the AMOC to the alternative state is possible in our model for $c_1 < c_{1_{crit}}$ and $c_2 < c_{2_{crit}}$ before their respective intrinsic tipping points are crossed (figure 6, moving from lower left along the pink arrow). In contrast to the previous example 3.1, the additional negative coupling results in the tipping of both interacting subsystems at an effective tipping point below their intrinsic tipping points. In a master–slave system (example 3.1), the master system $X_1$ needs to tip through an increase of its control parameter above its intrinsic tipping point $c_1 > c_{1_{crit}}$ to trigger a critical transition in the slave system $X_2$ at an effective tipping point $c_2 < c_{2_{crit}}$.

With increasing coupling strengths $d_{21}$ and $d_{12}$ the system behaviour changes. The system has one unstable fixed point and Kadyrov oscillations [85] occur for a wide range of the control parameters in the considered part of the ($c_1$, $c_2$)–parameter space (upper left of figure 4).

**Example 3.3.** Master–slave–slave system, e.g. propagation of critical transitions in lake chains

Consider a system consisting of three ($n = 3$) unidirectionally coupled tipping elements given by equations (2.2) and (2.3) with $i = 1, 2, 3$; $d_{12}, d_{23} > 0$ and $d_{21}, d_{32}, d_{31}, d_{13} = 0$. This type of coupling corresponds to the behaviour of a lake chain subject to an external input of nutrients as a control parameter (as e.g. in [57]). As in example 3.1, it is assumed that the lakes are connected through a unidirectional water stream [57,58,67,91]. The behaviour of subsystem $X_1$ corresponds to the behaviour of an uncoupled tipping element. Therefore, the eutrophication of the first lake in the lake chain, i.e. the tipping of subsystem $X_1$, is possible with an increase of its control parameter $c_1 > c_{1_{crit}}$. For $c_1 > c_{1_{crit}}$ only stable fixed points with subsystem $X_1$ in the alternative state exist. Additionally increasing the control parameters $c_2$, $c_3$ or both results in the loss of further stable fixed points and allows for critical transitions in the slave systems $X_2$ and $X_3$ (figure 7). In the following, the system

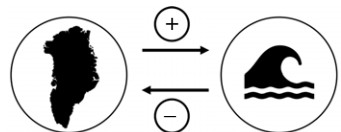

**Figure 6.** Number of stable fixed points and phase space portraits for two bidirectionally coupled tipping elements with $d_{21} = -0.2 < 0$ and $d_{12} = 0.2 > 0$ and low coupling strengths where $|d_{21}| = |d_{12}|$ depending on the control parameters $c_1$ and $c_2$. The dashed lines represent the intrinsic tipping point of the respective subsystem. The phase space portraits allow to derive the possible critical transitions in the master–slave system. Within the phase portraits stable fixed points are shown in orange, while unstable fixed points are shown in red. The background colour indicates the normalized speed $v = \sqrt{\dot{x}_1^2 + \dot{x}_2^2}/v_{max}$ going from close to zero (purple) to fast (yellow–green).

behaviour with $c_1 > c_{1_{crit}}$ for low coupling strengths $d_{12}, d_{23} > 0$ is analysed (see figure 7, lower left and electronic supplementary material, figure S3 for a zoom-in).

With an increasing control parameter $c_2$ or $c_3$, a critical transition in the corresponding subsystems is possible in our model for $c_2 < c_{2_{crit}}$ (electronic supplementary material, figure S4) or $c_3 < c_{3_{crit}}$ (electronic supplementary material, figure S5), given that the preceding subsystem occupies the alternative state or undergoes a transition into the alternative state through a continuously changing control parameter. Lake

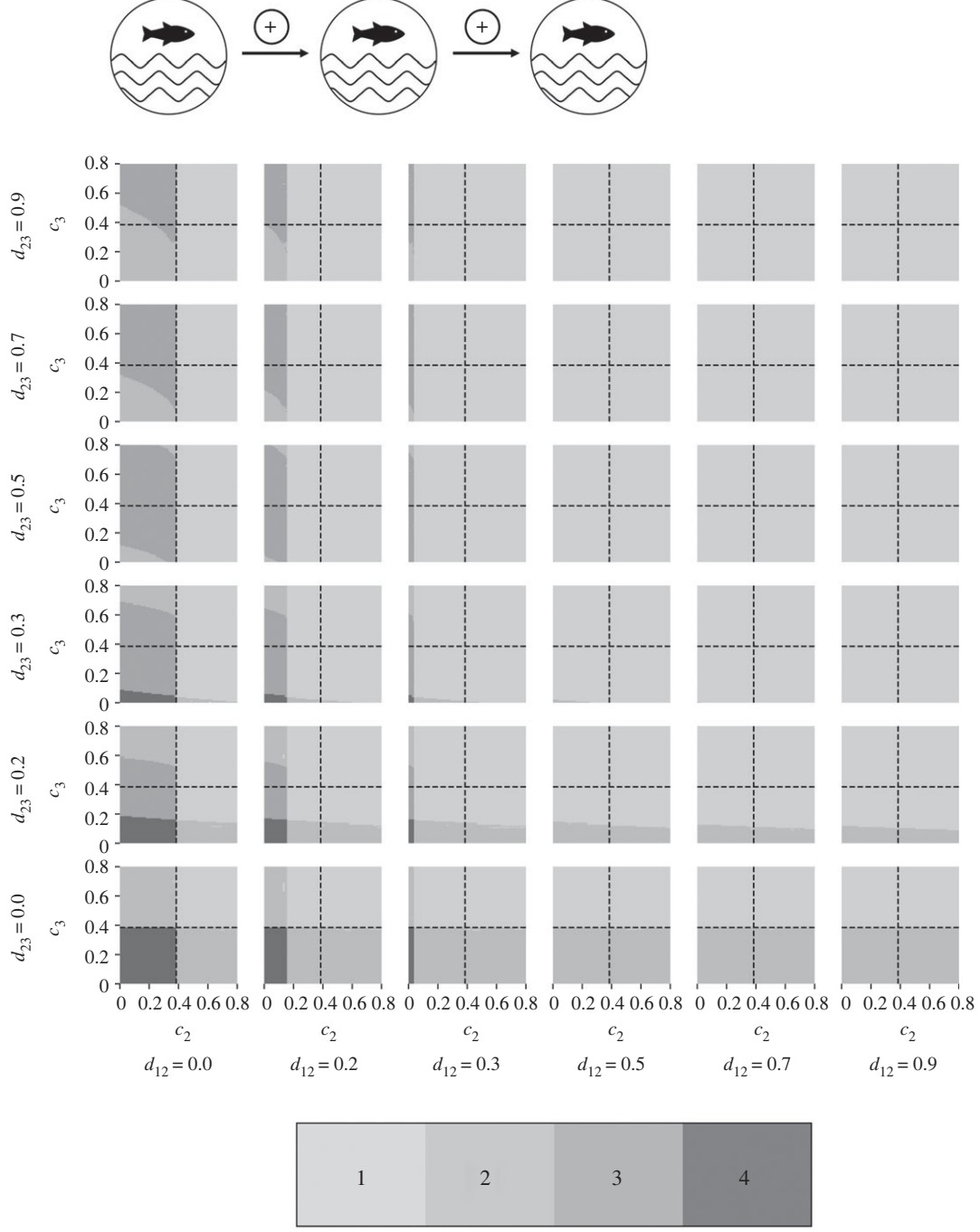

**Figure 7.** Number of stable fixed points of the system consisting of three unidirectionally coupled tipping elements for $c_1 = 0.4 > c_{1_{\text{crit}}}$ depending on the control parameters $c_2$ and $c_3$ and the coupling strengths $d_{12} \geq 0$ and $d_{23} \geq 0$ in a matrix of stability maps. A stability map shows the number of stable fixed points in the $(c_2, c_3)$–space for a specific coupling strength, where a certain number of stable fixed points is associated with a specific colour. Note that different areas in the control parameter space with the same colour have the same number of stable fixed points but they do not necessarily have the same phase portrait. The dashed lines represent the intrinsic tipping point of the respective subsystem. The position of a stability map in the matrix is determined by the coupling strength.

$X_2$ and $X_3$ in the lake chain can, therefore, become eutrophic before the intrinsic critical level of nutrient input of an isolated lake is crossed, given that the preceding lake has already become eutrophic.

For a simultaneous, slight increase of the control parameters $c_2$ and $c_3$ of both subsystems $X_2$ and $X_3$, a critical transition in both subsystems $X_2$ and $X_3$ is possible in our model for $c_2 < c_{2_{\text{crit}}}$ and $c_3 < c_{3_{\text{crit}}}$. As a result, a tipping cascade can be observed, given subsystem $X_1$ has already tipped or tips with $c_1 > c_{1_{\text{crit}}}$

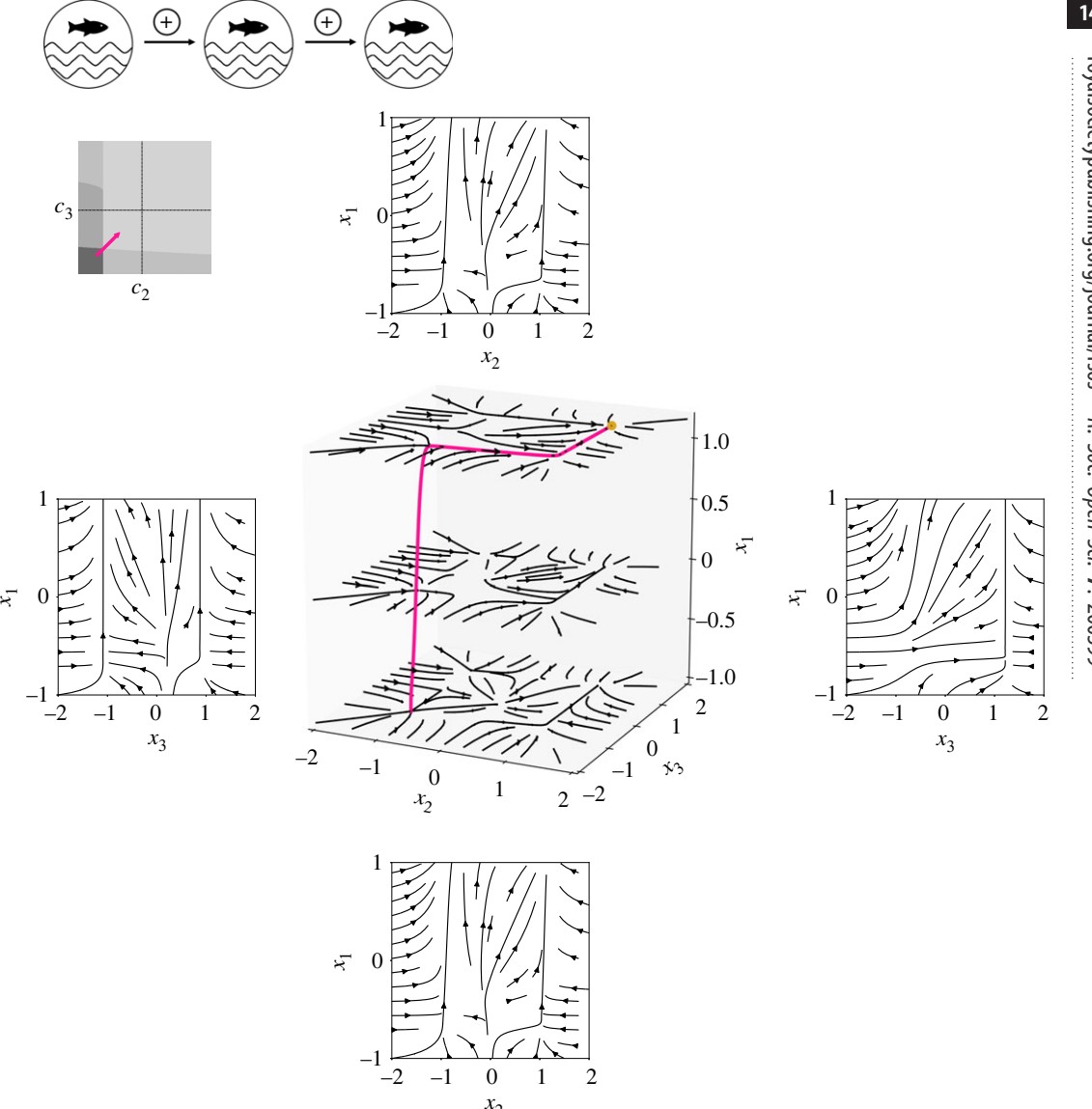

**Figure 8.** Tipping cascade in a system of three unidirectionally coupled tipping elements for an increase of the control parameters $c_2 << c_{2_{crit}}$ and $c_3 << c_{3_{crit}}$ and $c_1 > c_{1_{crit}}$ (as indicated by the pink arrow in the stability map, upper left panel). The central cube shows the flow in the $(x_2, x_3)$-space as part of the three-dimensional phase space and the remaining stable fixed point (in orange) for $c_1 = 0.4 > c_{1_{crit}}$, $c_2 = c_3 = 0.2$ with $d_{12} = 0.2 > 0$ and $d_{23} = 0.2 > 0$. The tipping cascade is highlighted by the pink trajectory. Two-dimensional plots arranged around the central cube show the flow in the $(x_2, x_1)$- and $(x_3, x_1)$-space corresponding to the lateral surfaces of the cube.

(pink trajectory in figure 8). Consequently, after the eutrophication of the first lake, a critical transition to the turbid state of a lake can spread in the lake chain even if the intrinsic critical value of nutrient input known from an isolated lake is not crossed.

# 4. Discussions and conclusion

The qualitative long-term behaviour of interacting, cusp-like tipping elements has been assessed in a simple analytic and an extensive numerical analysis of a conceptual model. Depending on the type of coupling and the coupling strength, qualitatively different behaviours of the systems of interacting tipping elements were observed. In particular, tipping cascades where a critical transition in one subsystem triggers the tipping of a coupled subsystem may occur under certain conditions.

Simple analytic calculations resulted in the formulation of tipping rules for the spread of tipping processes in systems of interacting tipping elements: a shift in the threshold value of the control parameter, at which an interacting tipping element undergoes a transition into a qualitatively different state, can occur. We call the threshold value of the isolated subsystem the intrinsic tipping point of the tipping element. If an interaction with another tipping element exists, the tipping process takes place when the so-called effective tipping point is crossed. Depending on the coupling direction, the effective tipping point can occur at either lower (facilitated tipping) or higher (impeded tipping) values of the control parameter than the intrinsic tipping point.

We have generalized and extended existing studies of special cases of coupled cusp-like tipping elements [52,85] through an extensive numerical analysis of two and three interacting tipping elements with one- or bidirectional coupling of varying direction. The behaviour of the special cases including a window with both subsystems of a simple master–slave system [52] in the alternative state, a tipping cascade in a positively coupled master–slave–slave system [52] and the Kadyrov oscillator [85] for two bidirectionally coupled tipping elements for a high coupling strength of same magnitude but with opposite signs is consistent with the system behaviour observed in our analysis.

In addition, our extensive analysis allowed to identify types of coupling that favour critical tipping scenarios. Conditions in terms of coupling strength and control parameters of the subsystem under which the tipping scenarios occur have been determined. Cascades of tipping processes that occur before the crossing of intrinsic tipping points, i.e. where the effective tipping point lies at lower values than the intrinsic tipping point of the uncoupled tipping element, are of special interest. In a simple master–slave system with positive coupling, a critical transition in the master system due to a crossing of its intrinsic tipping point triggers a critical transition of the slave system at an effective tipping point lower than its intrinsic tipping point. By contrast, a negative coupling would prevent a facilitated tipping of the slave system in the case of a master system being in the alternative state. In a system of two tipping elements with bidirectional coupling, a tipping cascade is favoured if one of the coupling terms is negative. In a master–slave–slave system with $d_{12} > 0$ and $d_{23} > 0$, the initial tipping of the master system can trigger cascading tipping processes in the following subsystems before the intrinsic threshold of the corresponding control parameter is crossed. Such a tipping cascade before the crossing of the corresponding intrinsic thresholds cannot be observed after the introduction of a negative coupling (results not shown here). Tipping processes are suppressed instead in this case and do not spread into all subsystems.

Applying the qualitative system behaviour to selected interacting real-world tipping elements revealed possible tipping scenarios, which are relevant for the future development of the Earth system, and in addition, due to the consequences of tipping, such as sea-level rise [92,93], for the economy, infrastructure and society more broadly. In particular, the analysis of the qualitative long-term system behaviour of two bidirectionally coupled tipping elements with opposite sign but same magnitude suggests that the Greenland ice sheet and the AMOC might tip before their intrinsic tipping points are reached. In other words, the meltdown of the Greenland ice sheet and the slowdown of the AMOC might begin before the intrinsic threshold ranges identified for isolated tipping elements [94] is crossed. The possible existence of such tipping cascades increases the risks that anthropogenic climate change poses to human societies, since the intrinsic threshold ranges of some climatic tipping elements including the Greenland ice sheet are assumed to lie even within the 1.5–2°C target range of the Paris agreement [94].

When it comes to the application of tipping behaviour to real-world systems, it should be noted that tipping elements were described in an idealized way using the normal form of the cusp catastrophe onto which, by the concept of topological equivalence [88], the critical behaviour of a class of real-world systems can be mapped. The proposed model of interacting tipping elements, therefore, shows a hypothetical, but mathematically possible system behaviour. It was motivated by its catastrophic features [46,87] in contrast to other bifurcational systems allowing the transition into a qualitatively different state by the variation of a bifurcation parameter and the appearance of the double fold bifurcation in many real world systems [6,8,12,15,17–19,95]. However, processes which are not taken into account in the conceptual representation of tipping elements, but are present in the real world, might influence the system and its tipping behaviour. In addition to a direct coupling of tipping elements in the climate system [54], an indirect, 'diffusive' interaction through e.g. the global mean temperature [1] could be considered as a potential coupling mechanism. Furthermore, chains and pairs of tipping elements have been analysed isolated from the larger network of interacting climatic tipping elements [54], i.e. possible interactions with other climatic tipping elements have been neglected. Here, we focused on bifurcation-induced tipping assuming that the control parameter

varies sufficiently slowly for the system to keep track with the stable states. It should be noted that a change of the control parameter with a high rate is likely, given the increasing influence of humans on the Earth system, possibly giving rise to rate-induced tipping [45,50].

The qualitative and theoretically possible system behaviours studied here and their application to real-world systems, therefore, introduces further research questions regarding tipping elements and their interactions in ecology, climate science and other fields. The conceptual approach should be extended to networks of tipping elements as already suggested in [84] and motivated in [54]. Networks of interacting tipping elements can be analysed using methods of statistical mechanics [96]. Critical transitions may spread across a whole network of tipping elements depending on the clustering and the spatial organization of the network [76–78]. Taking the important interactions of climatic tipping elements into account in a network approach, realistic complex models must be used for quantitatively approximating the effective tipping point. In addition, interacting tipping elements with heterogeneous intrinsic threshold and varying internal time scales should be considered [65] as, for example, the critical nutrient input of lakes varies with their depth [7]. Finally, generic early warning signals for tipping cascades comparable to already existing indicators for critical transitions of isolated tipping elements [97,98] are desirable to forecast cascading tipping events and counteract undesired consequences of tipping (cascades). A first step towards early warning indicators of tipping cascades has been presented only recently [86]. However, up to now, it remains an open question whether early warning signals for tipping cascades based on critical slowing down [97,98] exist.

Data accessibility. Data and relevant code for this research work are stored in GitHub: https://github.com/pik-copan/pytippinginteractions and have been archived within the Zenodo repository: https://doi.org/10.5281/zenodo.3768371.
Authors' contributions. A.K.K. and V.K. performed mathematical and numerical analysis. A.K.K. led the writing of the manuscript and created figures and tables. J.F.D. and R.W. conceived of the study, designed the study and coordinated the study. All authors helped in drafting the manuscript and critically revised the text. All authors gave final approval for publication and agree to be held accountable for the work performed therein.
Competing interests. We declare we have no competing interest.
Funding. V.K. thanks the German National Academic Foundation (Studienstiftung des deutschen Volkes) for financial support. J.F.D. is grateful for financial support by the Stordalen Foundation via the Planetary Boundary Research Network (PB.net), the Earth League's EarthDoc program and the European Research Council Advanced Grant project ERA (Earth Resilience in the Anthropocene). We are thankful for support by the Leibniz Association (project DominoES).
Acknowledgements. This work has been performed in the context of the copan collaboration and the FutureLab on Earth Resilience in the Anthropocene at the Potsdam Institute for Climate Impact Research. Furthermore, we acknowledge discussions with and helpful comments by N. Wunderling, J. Heitzig and M. Wiedermann.

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
