## [Reviewer comments · Royal Society Open Science]

Review History

Decision letter (RSOS-200599.R0)

Dear Dr Donges:

It is a pleasure to accept your manuscript entitled "Emergence of cascading dynamics in interacting tipping elements in ecology and climate" in its current form for publication in Royal Society Open Science.

Your manuscript was reviewed previously when it was submitted to the Journal of the Royal Society Interface and, as you know, those reviews were available in the consideration of the revised paper for Royal Society Open Science. The recommendation of the Associate Editor assigned to this paper was that you had responded very comprehensively to those reviews and that it was therefore appropriate to accept the paper for publication without further referee reports. I have accepted that recommendation.

Please ensure that you send to the editorial office an editable version of your accepted manuscript, and individual files for each figure and table included in your manuscript. You can send these in a zip folder if more convenient. Failure to provide these files may delay the

processing of your proof. You may disregard this request if you have already provided these files to the editorial office.

on behalf of Professor Matjaz Perc (Associate Editor) and Professor Peter Haynes (Subject Editor).

Associate Editor Professor Matjaz Perc Comments to Author:

Comments to the Author:

Thank you for sending us your revised manuscript for publication to Royal Society Open Science. We appreciate the care and love to detail that you have invested in responding to the previous Referee reports, and for the comprehensive revision of your manuscript. We are fully satisfied with the work you have done, and we are therefore happy to accept your manuscript for publication in present form. Congratulations to a fine contribution, and many thanks for choosing Royal Society Open Science to publish your research.
